# Immediate fluid management of children with severe febrile illness and signs of impaired circulation in low-income settings: a contextualised systematic review

Newton Opiyo,[1] Elizabeth Molyneux,[2] David Sinclair,[3] Paul Garner,[3] Mike English[1,4]

▶ Prepublication history and additional material is available. To view please visit the journal (http://dx.doi.org/10.1136/bmjopen-2014-004934).

For numbered affiliations see end of article.

**Correspondence to**
Dr Newton Opiyo; nopiyo@kemri-wellcome.org

## ABSTRACT

**Objective:** To evaluate the effects of intravenous fluid bolus compared to maintenance intravenous fluids alone as part of immediate emergency care in children with severe febrile illness and signs of impaired circulation in low-income settings.

**Design:** Systematic review of randomised controlled trials (RCTs), and observational studies, including retrospective analyses, that compare fluid bolus regimens with maintenance fluids alone. The primary outcome measure was predischarge mortality.

**Data sources and synthesis:** We searched PubMed, The Cochrane Library (to January 2014), with complementary earlier searches on, Google Scholar and Clinical Trial Registries (to March 2013). As studies used different clinical signs to define impaired circulation we classified patients into those with signs of severely impaired circulation, or those with any signs of impaired circulation. The quality of evidence for each outcome was appraised using the Grading of Recommendations Assessment, Development and Evaluation (GRADE) approach. Findings are presented as risk ratios (RRs) with 95% CIs.

**Results:** Six studies were included. Two were RCTs, one large trial (n=3141 children) from a low-income country and a smaller trial from a middle-income country. The remaining studies were from middle-income or high-income settings, observational, and with few participants (34–187 children).

**Severely impaired circulation:** The large RCT included a small subgroup with severely impaired circulation. There were more deaths in those receiving bolus fluids (20–40 mL/kg/h, saline or albumin) compared to maintenance fluids (2.5–4 mL/kg/h; RR 2.40, 95% CI 0.84 to 6.88, p=0.054, 65 participants, low quality evidence). Three additional observational studies, all at high risk of confounding, found mixed effects on mortality (very low quality evidence).

**Any signs of impaired circulation:** The large RCT included children with signs of both severely and non-severely impaired circulation. Overall, bolus fluids increased 48 h mortality compared to maintenance fluids with an additional 3 deaths per 100 children treated (RR 1.45, 95% CI 1.13 to 1.86, 3141 participants, high quality evidence). In a second small

### Strengths and limitations of this study

- Timely systematic review given the current uncertainty on optimal strategy for fluid resuscitation in children; review incorporates data on the largest randomised controlled trial (RCT; Fluid Expansion as Supportive Therapy trial) of fluid therapy in children.
- Review includes all relevant comparative trials and observational data, and critically appraises the research evidence using Grading of Recommendations Assessment, Development and Evaluation (GRADE) methods.
- Review is limited because most studies are small and unreliable, with only one large RCT providing reliable data to guide policy.

RCT from India, no difference in 72 h mortality was detected between children who received 20–40 mL/kg Ringers lactate over 15 min and those who received 20 mL over 20 min up to a maximum of 60 mL/kg over 1 h (147 participants, low quality evidence). In one additional observational study, resuscitation consistent with Advanced Paediatric Life Support (APLS) guidelines, including fluids, was not associated with reduced mortality in the small subgroup with septic shock (very low quality evidence).

**Signs of impaired circulation, but not severely impaired:** Only the large RCT allowed an analysis for children with some signs of impaired circulation who would not meet the criteria for severe impairment. Bolus fluids increased 48 h mortality compared to maintenance alone (RR 1.36, 95% CI 1.05 to 1.76, high quality evidence).

**Conclusions:** Prior to the publication of the large RCT, the global evidence base for bolus fluid therapy in children with severe febrile illness and signs of impaired circulation was of very low quality. This large study provides robust evidence that in low-income settings fluid boluses increase mortality in children with severe febrile illness and impaired circulation, and this increased risk is consistent across children with severe and less severe circulatory impairment.

## BACKGROUND

Health staff must rapidly assess, resuscitate and treat severely ill children to improve survival. Practical, simple to use protocols have been established to guide care. In North America, the most commonly used are the Pediatric Advanced Life Support (PALS)[1] and in Europe, the European Paediatric Life Support (EPLS)[2] guidelines. In low-income settings, the WHO has specific guidelines where there are no intensive care facilities, called the Emergency Triage Assessment and Treatment (ETAT) guidelines. ETAT guidance begins with initial triage to identify children who need urgent formal assessment such as altered consciousness or severe breathing or circulation problems. Immediate care should then follow an Airway, Breathing, Circulation, Drugs (ABCD) approach to assessment and action which prioritises care of the airway (A) first, followed by breathing (B) and circulation (C).

### Septic shock

Severe febrile illness in children is often associated with signs of impaired circulation (also known as 'septic shock'). Current ETAT guidelines define shock as the presence of *three* clinical signs of poor peripheral perfusion: weak/absent peripheral pulse, prolonged capillary refilling >3 s *and* cold hands and feet (typically with cold skin extending up the limb and termed 'a temperature gradient'). International guidance, including ETAT, typically recommends a rapid fluid bolus of 20–40 mL/kg intravenously once 'shock' is diagnosed.[3]

### Why it is important to do this review

As early as 1999, authors were arguing that no robust data existed demonstrating that bolus fluid resuscitation improved clinical outcomes in septic patients.[4] Moreover, the results of observational studies suggested that large volumes of resuscitation fluids may be associated with increased morbidity in patients with sepsis[5] acute respiratory distress syndrome[6] or acute kidney injury.[7]

Subsequently, a large, multicentre, randomised controlled trial (RCT) published in 2011, and conducted in East Africa found that bolus fluid resuscitation increased mortality compared to a maintenance fluid regimen in children with 'febrile illness' and impaired perfusion.[8] The results of this trial clashed with the long-standing belief that fluid bolus resuscitation was beneficial and caused considerable international debate.[9–12] To date, no clear guidance has emerged on how to incorporate these findings into recommendations for practice in high-income or low-income settings.

## OBJECTIVE

To evaluate the effects of fluid bolus (with either colloids or crystalloids) compared to maintenance fluids alone in children with severe febrile illness and signs of impaired circulation.

## METHODS

### Criteria for considering studies for this review

#### Types of studies

RCTs and observational studies, including retrospective analyses.

#### Types of participants

Children (aged ≤18 years) with clinical features suggesting impaired perfusion (including shock) due to presumed acute infectious illnesses or inflammatory state (sepsis) and excluding diarrhoeal illness (as defined by the studies).

#### Types of interventions

Intravenous fluid boluses (crystalloids or colloids) compared to no (or lower volume) fluid boluses or maintenance fluids.

#### Types of outcome measures

The primary outcome measure was predischarge mortality. Secondary outcomes were: mortality at any time up to 4 weeks and any adverse clinical events reported in the studies.

### Data sources and search strategy

We searched PubMed, The Cochrane Library to January 2014, complemented by searches in Google Scholar and Clinical Trial Registries (ClinicalTrials.gov, Current Controlled Trials, WHO International Clinical Trials Registry Platform, metaRegister of Controlled Trials) to March 2013. We sought eligible published, unpublished or in-progress articles. No date or language restrictions were used.

The searches were performed iteratively by combining Medical Subject Headings (MeSH) and free-text terms relevant to the conditions (sepsis, septicaemia, febrile illness, bacteraemia, infection, meningitis, septic shock, hypovolaemia), treatments (fluids, resuscitation, intravenous fluids, fluid therapy) and patient groups (neonates, infants, children, adolescents) of interest. No date or language restrictions were used.

In addition we searched the websites of relevant organisations (the International Sepsis Forum, the World Federation of Paediatric Intensive Care and Critical Care Societies) and key emergency/intensive care journals (Critical Care Medicine, Critical Care, Paediatric Emergency Medicine, Shock, Resuscitation, Intensive Care Medicine). Reference lists of related systematic reviews and primary studies were manually searched. We also sought additional papers by contacting authors of related reviews and selected studies.

### Study selection

Two reviewers (NO and ME) independently screened the titles, abstracts and full texts of retrieved articles and applied the study eligibility criteria detailed above to select studies. Disagreements were resolved by discussion.

## Data extraction

Data were extracted using a predesigned form by one reviewer (NO) and checked by the other reviewers; disagreements were resolved by discussion. We extracted data on: study designs, settings, sample size, participants (diagnoses, age range), shock definitions, treatments and comparisons (types of fluids, timing, volumes and fluid rates), cointerventions and proportion of patients experiencing the events of interest in each treatment group.

## Risk of bias in individual studies

Two reviewers (NO and DS) independently assessed the risk of bias in the included studies according to six criteria assessing the risk of selection bias (random sequence generation, allocation concealment, selection of two groups), reporting bias (blinding) and confounding (baseline characteristics, cointerventions). For each criteria, the study was classified as high risk of bias, low risk of bias or unclear risk of bias.

## Assessment of quality of evidence

The quality of evidence for each of the efficacy and safety outcomes was assessed using the Grading of Recommendations Assessment, Development and Evaluation (GRADE) approach.[13] Key quality elements assessed by GRADE include: risk of bias (study limitations), precision of treatment effects, consistency of results, directness (applicability) of evidence and publication bias. The GRADE evidence profiles were prepared by one reviewer (NO) and verified independently by two reviewers (ME and DS). Discrepancies in the quality ratings were resolved by discussion.

## Synthesis of results

We summarised results narratively due to significant differences in study designs, fluid protocols and patient risk profiles. In order to compare studies with similar populations we grouped studies by the severity of circulatory impairment at baseline, after an appraisal of the clinical signs used to define inclusion. We defined three groups (table 1):

1. *Severely impaired circulation (SIC)*: Studies where inclusion criteria were similar to the ETAT guidance (shock defined as presence of all four signs of impaired circulation).
2. *Impaired circulation (IC):* Studies where inclusion only required one or two of these signs.
3. *IC but without severe impairment:* Studies where inclusion required one or two signs of IC and more severely ill patients were excluded.

For consistency all results are presented as risk ratios (RRs) with 95% CIs (where reported, odds ratios and percentage point differences were converted into RRs).

## RESULTS

### Study selection process

The flow of studies through this review is summarised in figure 1. Six studies fulfilled all our prespecified eligibility criteria: two RCTs,[8 14] two prospective cohort studies[15 16] and two retrospective record reviews.[17 18]

### Study characteristics

The characteristics of the six included studies are summarised in online supplementary table S1. Study settings were varied: USA,[15–17] Brazil,[18] India[14] and East Africa (Kenya, Tanzania and Uganda).[8] Five studies[14–18] were conducted in settings where paediatric intensive care unit facilities including inotropic support, intubation and ventilation were available. The largest study[8] was conducted in typical resource-limited East African hospitals without these additional measures being available. The study sample sizes ranged from 34 to 3141 patients.

The clinical definitions of severe febrile illness and circulatory impairment or 'shock' varied across studies

**Table 1** Severity of circulatory impairment classifications

| Clinical group | Definition |
|---|---|
| SIC | Children with severe febrile illness who have *all four* of the following features: |
| | ▶ AVPU<A |
| | ▶ Weak/absent peripheral pulse |
| | ▶ Prolonged capillary refilling >3 s |
| | ▶ Cold limb extremities (hands and feet) typically with cold skin extending up the limb (referred to as a temperature gradient) |
| | These children typically also have secondary signs such as altered consciousness |
| IC | Children with severe febrile illness who may have *AVPU<A, or prostration or respiratory distress plus at least ONE* of the following features are included in this group: |
| | ▶ Weak peripheral pulse |
| | ▶ Capillary refilling >2 s |
| | ▶ Cold limb extremities with a temperature gradient |
| | ▶ Severe tachycardia (>180/min if aged 2–12 m, >160/min if aged 1–4 years) |
| IC but without SIC | By exclusion a third clinical grouping can be defined, those with impaired circulation but without severe impairment |

AVPU, Alert, responsive to Verbal, Painful stimuli, or Unresponsive; IC, impaired circulation; SIC, severely impaired circulation.

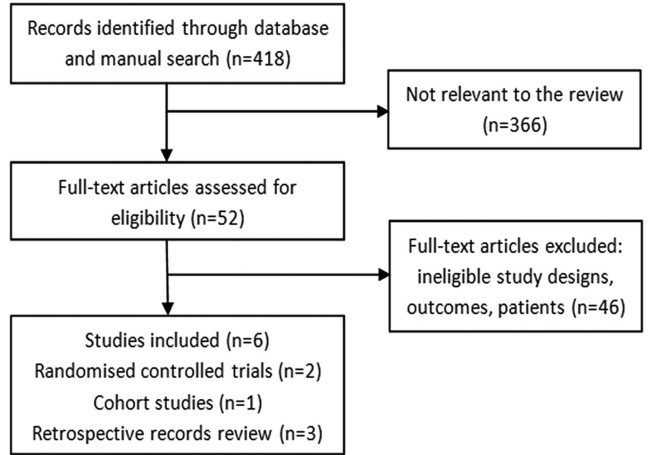

Records identified through database and manual search (n=418)

Not relevant to the review (n=366)

Full-text articles assessed for eligibility (n=52)

Full-text articles excluded: ineligible study designs, outcomes, patients (n=46)

Studies included (n=6)
Randomised controlled trials (n=2)
Cohort studies (n=1)
Retrospective records review (n=3)

**Figure 1** Results of literature search and studies selected.

(table 2). Fluid therapy protocols (volumes, timing, rates) were similarly varied between comparison study groups (see online supplementary table S1). In only one case, the work conducted in East Africa, was a clearly defined 'no bolus' group studied.[8]

## Risk of bias

Both RCTs were considered to be of low risk of bias (table 3). The observational studies were all potentially confounded (none met all the risk of bias criteria; table 3). The following key criteria were only partially met or were unclear from the information provided: appropriate participant selection, study power, appropriate blinding/outcome assessments and association of outcome and treatment.

## Outcomes

The results for the efficacy and safety outcomes, together with quality of evidence, are summarised in the following three distinct population risk groups.
A. Children with severe febrile illness and signs of SIC

Four studies reported outcome data in children relevant to those with SIC: one RCT,[8] one prospective cohort[16] and two retrospective studies.[17] [18] Overall mortality in the patients reported in these analyses were: 42%, 51%, 29% and 47%, respectively.

*Trial data*: The RCT[8] included children with milder forms of circulatory impairment but provided a subgroup analysis of 65 patients who fulfilled the ETAT criteria of 'shock'. In this subgroup a fluid bolus of 20–40 mL/kg (saline or albumin) over 1 h was associated with a considerably higher risk of mortality than maintenance fluids (2–4 mL/kg/h) but this was of borderline statistical significance (RR 2.40, 95% CI 0.84 to 6.88, p=0.054, *low quality evidence*; see online supplementary tables S2 and S3).

*Observational data*: In Brazil a retrospective records review of 90 children admitted to a paediatric intensive care unit with sepsis and shock assessed the relationship between mortality and fluid resuscitation in the first

hour.[18] Administration of 20 mL/kg of bolus resuscitation fluid (crystalloids, colloids) was associated with a significantly higher mortality than if 40 mL/kg or more was given (73% vs 33%, RR 0.45, 95% CI not estimable, p<0.05, *very low quality evidence*; see online supplementary tables S2 and S3). Of particular note 80 of the 90 children studied had severe, pre-existing chronic disease such as malignancy.

A second retrospective records review from the USA, examined mortality in 91 children being resuscitated and referred by community physicians.[17] In this study 'appropriate fluid therapy' was reported to be strongly associated with improved survival (RR 0.20, 95% CI 0.05 to 0.78, *very low quality evidence*; see online supplementary tables S2 and S3). However, the 'appropriate fluid therapy' group included those with fluids given as per guidelines and children in whom signs of shock resolved quickly irrespective of the volumes of fluid given. In fact, the median fluid volumes given to children who died (32.9 mL/kg) were higher than those given to survivors (20 mL/kg).

The third study, a prospective cohort from the USA, included 34 children admitted to the paediatric intensive care unit with microbiologically proven septic shock, and all were receiving inotropes and had a pulmonary catheter inserted.[16] Those receiving less than 20 mL/kg of fluids during the first hour of resuscitation (normal saline, Ringers lactate, 5% albumin) had significantly higher mortality than those given 20–40 mL/kg of fluids (RR 1.11, 95% CI 0.59 to 2.11, *very low quality evidence*) or more than 40 mL/kg (RR 0.19, 95% CI 0.03 to 1.30, *very low quality evidence*; see online supplementary tables S2 and S3).
B. Children with severe febrile illness and any sign of IC

Two RCTs[8] [14] and one prospective cohort study[15] enrolled febrile children with clinical signs sufficiently similar to be included in the IC category. Overall mortality in the patients reported in these studies were 9.5%, 18% and 12.5%, respectively.

*Trial data*: In the largest RCT from Africa 3141 children were randomised to a fluid bolus (20–40 mL/kg albumin or normal saline over 1 h) or maintenance fluids (2.5–4.0 mL/kg/h).[8] Compared to maintenance fluids, the fluid bolus was associated with increased 48 h mortality (RR 1.45, 95% CI 1.13 to 1.86, *high quality evidence*; see online supplementary tables S4 and S5) and increased mortality at 4 weeks (RR 1.39, 95% CI 1.11 to 1.74, *moderate quality evidence*; see online supplementary tables S4 and S5). There was no suggestion that albumin performed any differently than saline (albumin bolus vs saline bolus, RR 1.0, 95% CI 0.78 to 1.29). There was no difference between the bolus and maintenance fluid groups in the risk of neurological sequelae at 4 weeks (RR 1.03, 95% CI 0.61 to 1.75, *low quality evidence*; see online supplementary tables S4 and S5) or the combined outcome of pulmonary oedema or increased intracranial pressure (RR 1.46, 95% CI 0.85 to 2.53, *low quality evidence*; see online supplementary tables S4 and

**Table 2** Study inclusion criteria

| | Study population entry criteria | | | 'Shock' criteria | | | | | | | | | | | | |
|---|---|---|---|---|---|---|---|---|---|---|---|---|---|---|---|---|
| Study | Age range | Severe illness | | Blood pressure | | Pulse rate | | Capillary refill | | Extremities | | Peripheral pulse | | Urine output | | Mental status |
| Maitland 2011 | 60 days to 12 years | Severe febrile illness complicated by impaired consciousness (prostration or coma), respiratory distress (increased work of breathing) or both | and | – | or | Severe tachycardia* | or | ≥3 s | or | Lower limb temperature gradient | or | Weak radial pulse volume | – | – | – | – |
| Oliveira 2008 | Median age: 36–47 months | Sepsis was defined using the Society of Critical Care Medicine Consensus Conference[22] | and | <5th centile for age | or | – | or | <1 s or >3 s | or | Mottled/cool | or | Decreased | or | <1 mL/kg/h | or | Altered |
| Han 2003 | 1–131 months | Suspected infection as manifested by hyperthermia or hypothermia | and | <5th centile for age | or | – | or | >3 s | or | Mottled | or | Diminished | or | – | | Decreased |
| Carcillo 1991 | Median age 13.5 months (range 1–192 months) | Sepsis was diagnosed if the patient had a positive blood culture or if a pathological organism from a tissue site was identified | and | <2 SD below mean | +3 of | Tachycardia† | or | – | or | Mottled/cool | or | Decreased | or | <1 mL/kg/h‡ | – | – |
| Santhanam 2008 | 1–12 months | Septic shock was defined | and | – | or | Tachycardia | | >2 s | or | Mottled/cool | or | Decreased | or | Decreased | or | Altered alertness |

Continued

Table 2 Continued

| | Study population entry criteria | 'Shock' criteria | | | | | |
|---|---|---|---|---|---|---|---|
| | using the Sepsis Consensus Conference criteria[23] | + 1 of | | | | | or Altered |
| Carcillo 2009 | Newborn to 18 years | Unclear from the information provided†† | <5th centile for age and | Tachycardia or | >3 s or | Mottled or | – or |

*>180 bpm in children younger than 12 months of age, >160 bpm in children 1–5 years of age, or 140 bpm in children older than 5 years of age.
†Heart rate >180 bpm for patients less than 5 years of age; and >160 bpm for patients at least 5 years of age.
††No definition of sepsis provided.
‡Or less than 20 mL/h in children weighing more than 20 kg.

S5). A subgroup analysis suggested that the increased mortality was only statistically significant in children with severe anaemia (haemoglobin <5 g/dL; RR 1.71, 95% CI 1.16 to 2.51, *moderate-quality evidence*; see online supplementary tables S4 and S5), but a subsequent analysis exploring the effect of anaemia when treated as a continuous variable found evidence of harm with fluid bolus across the full range of haemoglobin values.[19]

In the second RCT, 147 children in an Indian paediatric intensive care unit were randomised to receive Ringers lactate 20–40 mL/kg over 15 min or Ringers lactate 20 mL/kg over 20 min up to a maximum of 60 mL/kg over 1 h.[14] There was no difference in 72 h mortality between comparison groups (RR 0.99, 95% CI 0.49 to 1.98, *low quality evidence*; see online supplementary tables S4 and S5).

*Observational data*: The prospective cohort study included 1409 children but only 187 had septic shock and so were relevant to this review.[15] No difference in mortality was observed in this subgroup between those receiving resuscitation consistent with PALS/Advanced Paediatric Life Support recommendations (including rapid bolus 20 mL/kg of isotonic fluid, potentially repeated and use of inotropes) performed by community physicians compared to resuscitation not consistent with these recommendations (*very low quality evidence*; see online supplementary tables S4 and S5). In addition, the actual fluid volumes administered and the relationship between fluid volumes given and outcomes is not presented. Furthermore, resuscitation episodes were classified as having been consistent with guidelines if signs of shock resolved early in the course of intervention, irrespective of actual fluid volumes given.

C. Children with severe febrile illness and IC but not SIC

It was possible to derive outcome data for children with severe febrile illness and any sign of circulatory impairment (IC) but not SIC in one study[20]: 20–40 mL/kg bolus fluids (albumin, normal saline) provided over 1 h, compared to 2.5–4.0 mL/kg/h maintenance fluids, was associated in this large subgroup (n=3076 children) with increased 48-h mortality (RR 1.36, 95% CI 1.05 to 1.76, *high quality evidence*).

## DISCUSSION

This review was conducted to facilitate revision of national paediatric fluid management guidelines in Kenya and potentially neighbouring countries using ETAT guidance, but has direct policy implications for healthcare across Africa.

The limited data available prior to the Fluid Expansion as Supportive Therapy (FEAST) study demonstrate that the current recommendations for fluid bolus included in the ETAT guidelines were not supported by a strong scientific evidence base. In fact, the evidence was largely observational and unreliable. Added to this, the studies were themselves flawed, with

**Table 3** Quality assessment of included studies

| Study | Study design | Selection bias | | | Reporting bias | Confounding | |
| | | Random sequence generation | Allocation concealment | Selection of two groups | Blinding | Baseline characteristics | Co-interventions |
|---|---|---|---|---|---|---|---|
| Maitland et al[8] | Randomised controlled trial | Low risk | Low risk | Low risk | Low risk | Low risk | Low risk |
| Santhanam et al[14] | Randomised controlled trial | Low risk | Low risk | Low risk | Low risk | Low risk | Low risk |
| Carcillo et al[15] | Prospective cohort | NA | NA | High risk* | NA | High risk* | Unclear risk |
| Oliveira et al[18] | Retrospective records review | NA | NA | High risk† | NA | High risk† | Unclear risk |
| Han et al[17] | Retrospective records review | NA | NA | High risk‡ | NA | High risk‡ | Unclear risk |
| Carcillo et al[16] | Prospective cohort | NA | NA | Low risk | NA | Unclear risk§ | Unclear risk |

*Compared those who received recommended APLS/PALS treatment with those who did not. 'Received recommended APLS/PALS fluid therapy' was defined as those who recovered regardless of fluid therapy plus those who did not recover but received >20 mg/kg of fluids. Children who did not receive recommended APLS/PALS treatment were significantly younger and had significantly longer capillary refill times, lower blood pressure and higher oxygen requirements.
†Compared survivors and non-survivors. The survivors were more likely to have had higher fluid volumes and were also significantly younger.
‡Compared survivors and non-survivors. 'Appropriate fluid therapy' group includes those where fluid was given in line with ACCM/PALS guidelines AND those who recovered quickly irrespective of how much fluid was given. Non-survivors had significantly higher PRISM scores at baseline (PRISM assesses the risk of mortality).
§Compared those who received three different fluid regimens. Adequate baseline characteristics were not presented.
APLS, Advanced Paediatric Life Support; NA, not applicable; PALS, Pediatric Advanced Life support.

design aspects which potentially biased the result towards favouring high-volume fluid resuscitation. Of particular note are two of the retrospective studies which classified all children who recovered quickly as having received 'adequate fluid therapy' irrespective of the fluid volume they received. The potential for bias in these studies resulting from possible exposure misclassification should be noted by paediatricians working in high-resource settings.

The RCT published in 2011 provides by far the most direct assessment of fluid boluses in children with severe febrile illness in resource poor African settings that do not typically see dengue fever but where malaria may be common.[8] In these settings emergency management decisions must typically be made without accurate blood pressure reading or investigations such as pulse oximetry, blood gas analysis, haemoglobin or lactate measurement. The robust finding of increased mortality in the large group of children with an initial, clinical diagnosis of severe febrile illness and IC but not SIC demonstrates that the clinical signs linked to this classification are, alone, not sufficient to identify children in whom fluid boluses may be beneficial. In fact this trial provides robust evidence that bolus intravenous fluids are harmful in such children.

Data from the small subgroup of children within the FEAST study who had all three clinical signs of SIC, are the only randomised evidence on the risks and benefits of fluid boluses for this group. Mortality among those receiving bolus fluids was higher than those receiving maintenance fluids but the data are compatible with an effect ranging from a small potential benefit of bolus to very substantial harm, which raises severe doubt about the use of boluses even in this more severely ill group.

In an effort to accommodate the highly influential observational research on fluid use in paediatric emergency care we included data from studies traditionally excluded from systematic reviews of alternative therapies (such as the recent review by Ford et al[21]). Although this presented challenges it is an advantage of the GRADE approach that the quality of evidence from such studies can be transparently appraised and, potentially therefore, contribute to informed decision-making.

## CONCLUSION

Prior to the publication of the large multicentre African RCT, the evidence in support of aggressive fluid therapy in children with septic shock was only of very low quality. The 2011 RCT provides robust evidence that in low-resource settings fluid boluses, even in relatively small amounts, increase mortality in children with severe febrile illness and signs of IC. For children with signs of SIC the evidence suggests harm but there is less certainty and further research is warranted.

**Author affiliations**
[1]Health Services Unit, KEMRI-Wellcome Trust Research Programme, Nairobi, Kenya
[2]Department of Paediatrics, College of Medicine and Queen Elizabeth Central Hospital, Blantyre, Malawi
[3]International Health Group, Liverpool School of Tropical Medicine, Liverpool, UK

[4]Nuffield Department of Medicine and Department of Paediatrics, University of Oxford, Oxford, UK

**Contributors** NO conducted the searches. NO, DS and ME screened records for eligibility, extracted data, assessed study quality and analysed data. EM and PG reviewed selected studies. NO wrote the first draft of the manuscript. All the authors participated in the interpretation of results and writing of the full manuscript. NO and ME are the guarantors.

**Funding** NO is funded by a post-doctoral training grant from the Consortium for National Health Research. ME is supported by a Wellcome Trust Senior Fellowship (#097170). Additional funds from a Wellcome Trust Strategic Award (#084538) and a Wellcome Trust core grant awarded to the KEMRI-Wellcome Trust Research Programme (#092654) made this work possible. The guideline development meeting was supported in part by the Effective Health Care Research Consortium, which is funded by UKaid from the UK Government Department for International Development.

**Competing interests** EM worked with WHO to develop the initial ETAT training and supports the provision of ETAT training in Malawi and other countries. ME adapted ETAT when creating ETAT+ and supports the provision of ETAT+ training in Kenya and has facilitated use of ETAT+ in other countries.

**Provenance and peer review** Not commissioned; externally peer reviewed.

**Data sharing statement** No additional data are available.

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
