## [Reviewer comments · BMJ Open]

Some articles will have been accepted based in part or entirely on reviews undertaken for other BMJ Group journals. These will be reproduced where possible.

ARTICLE DETAILS

TITLE (PROVISIONAL)	Immediate fluid management of children with severe febrile illness and signs of impaired circulation in low income settings: a contextualized systematic review
AUTHORS	Opiyo, Newton; Molyneux, Elizabeth; Sinclair, Dave; Garner, Paul; English, Mike

VERSION 1 - REVIEW

REVIEWER	Mark E. Ralston MD MPH Department of Pediatrics Naval Hospital Oak Harbor, Washington USA Department of Pediatrics Uniformed Services University of the Health Sciences Bethesda, Maryland USA
REVIEW RETURNED	11-Mar-2014

GENERAL COMMENTS	This study highlights the findings of the large RCT from East Africa (FEAST Trial) which has significantly improved the evidence base regarding fluid management in children in low-income settings with severe infection (particularly P. falciparum infection) and impaired circulation. The study should revise its statement in the discussion section that the findings of the FEAST Trial important to East Africa can be extrapolated to affect “those involved in paediatric care in high-resource settings.” In these industrialized countries, children with septic shock likely present earlier and have relevant ventilatory, inotropic, and monitoring support within an ICU.
---

REVIEWER	Professor John Myburgh University of New South Wales and the George Institute for Global Health, Sydney, Australia I have no competing interests in relation to this manuscript. My institution, the George Institute for Global Health, has received unrestricted grant funding and travel reimbursements from Fresenius Kabi in relation to the completion of an investigator-initiated RCT, and monies from Baxter in relation to participation on Advisory Boards.
REVIEW RETURNED	12-Mar-2014

GENERAL COMMENTS	This is a well conducted systematic review of fluid resuscitation in
--

	children in low-income countries. This review was clearly prompted by the discussions and commentaries that followed the publication of the FEAST trial in 2011. This landmark, high quality randomised-controlled trial challenged fundamental and strongly-held perspectives on fluid resuscitation in children and drew substantial criticism primarily relating to the generalisability of its findings in high and middle-income countries. Unsurprisingly, many of the commentators have emanated from high income countries and include prominent members of guideline-writing committees. Despite the validity and probity of the FEAST study and the publication of subsequent analyses defining potential biological mechanisms associated with bolus-induced mortality in this population, practice guidelines, including those from the WHO continue to recommend boluses for fluid resuscitation in this population. The evidence on which these guidelines have been based are largely of low quality, compromising observational and cohort studies, all of which are subject to high levels of bias. This review is therefore timely and clearly defines the evidence base, both in terms of quantity and validity. The objectives, methodology and internal validity of this systematic review is of the highest quality and conforms to all criteria defined in the PRISMA process. The FEAST study dominates the dataset and analyses - this is an important outcome from the review and places the weighting of low-quality evidence into perspective. The results are presented in three pre-defined strata and the conclusions are consistent with the level of evidence. The discussion is appropriately conservative and there is little editorial comment, which is commendable given the high level of emotive commentary on this subject. The authors are to be commended on conducting this review.
--	--

REVIEWER	Eddy Lang University of Calgary, Calgary, Alberta, Canada Member GRADE working group
REVIEW RETURNED	17-Mar-2014

GENERAL COMMENTS	The authors present a systematic review focusing on the mortality effects of high volume versus maintenance levels of fluid resuscitation in children with sepsis who present in developing health care systems. The authors identify and include two RCTs with one being the dominant work in this area as well as a series of observational studies. The review addresses an important and controversial question and provides an important contribution to the medical literature. While the findings are driven by the Maitland trial there is still merit in exploring the impact of aggressive fluid intervention across other trials in the hopes of facilitating generalizability. In regards to methodology, the authors are adherent to PRISMA guidelines (except for providing a protocol or registering with PROSPERO) and do a good job applying the GRADE framework to rating the quality of evidence. Specific comments:
--

	Abstract: Creating sub-divisions in the result section is an uncommon format and consolidation into a unified results section would be preferable. Synthesis of results: This section is confusing as the three categories identified do not appear to be mutually exclusive with significant overlap between the second and third. Perhaps a table would be helpful here. Summary of findings tables (GRADE) Please note that these should be updated. The definitions as they pertain to quality of evidence are no longer valid. Evidence is conceptualized as confidence in estimates of effect and not in terms of the likelihood of future research making a difference. Table 4 Foot note 3 is problematic. The basis of the indirectness is unclear i.e. why is the African setting problematic? Also serious indirectness usually implies a drop by two levels. GRADE evidence profiles would be useful to include as well to facilitate a better understanding of the limitations of the evidence base.
--	--

VERSION 1 – AUTHOR RESPONSE

Reviewer: 1

This study highlights the findings of the large RCT from East Africa (FEAST Trial) which has significantly improved the evidence base regarding fluid management in children in low-income settings with severe infection (particularly *P. falciparum* infection) and impaired circulation.

The study should revise its statement in the discussion section that the findings of the FEAST Trial important to East Africa can be extrapolated to affect “those involved in paediatric care in high-resource settings.” In these industrialized countries, children with septic shock likely present earlier and have relevant ventilator, inotropic, and monitoring support within an ICU.

Authors’ responses

Agree: statement on the applicability of FEAST trial findings to high-resource settings now omitted.

Reviewer: 2

This is a well conducted systematic review of fluid resuscitation in children in low-income countries. This review was clearly prompted by the discussions and commentaries that followed the publication of the FEAST trial in 2011. This landmark, high quality randomised-controlled trial challenged fundamental and strongly-held perspectives on fluid resuscitation in children and drew substantial criticism primarily relating to the generalisability of its findings in high and middle-income countries.

Unsurprisingly, many of the commentators have emanated from high income countries and include prominent members of guideline-writing committees. Despite the validity and probity of the FEAST study and the publication of subsequent analyses defining potential biological mechanisms associated with bolus-induced mortality in this population, practice guidelines, including those from the WHO continue to recommend boluses for fluid resuscitation in this population.

The evidence on which these guidelines have been based largely of low quality, compromising observational and cohort studies, all of which are subject to high levels of bias. This review is therefore timely and clearly defines the evidence base, both in terms of quantity and validity. The

objectives, methodology and internal validity of this systematic review is of the highest quality and conforms to all criteria defined in the PRISMA process.

The FEAST study dominates the dataset and analyses - this is an important outcome from the review and places the weighting of low-quality evidence into perspective. The results are presented in three pre-defined strata and the conclusions are consistent with the level of evidence.

The discussion is appropriately conservative and there is little editorial comment, which is commendable given the high level of emotive commentary on this subject. The authors are to be commended on conducting this review.

Authors' responses

Agree with the reviewer's comments.

Reviewer: 3

The authors present a systematic review focusing on the mortality effects of high volume versus maintenance levels of fluid resuscitation in children with sepsis who present in developing health care systems. The authors identify and include two RCTs with one being the dominant work in this area as well as a series of observational studies.

The review addresses an important and controversial question and provides an important contribution to the medical literature. While the findings are driven by the Maitland trial there is still merit in exploring the impact of aggressive fluid intervention across other trials in the hopes of facilitating generalizability.

In regards to methodology, the authors are adherent to PRISMA guidelines (except for providing a protocol or registering with PROSPERO) and do a good job applying the GRADE framework to rating the quality of evidence.

Specific comments:

Abstract:

Creating sub-divisions in the result section is an uncommon format and consolidation into a unified results section would be preferable.

Authors' response

Agree, results now consolidated and presented in paragraphs.

Synthesis of results:

This section is confusing as the three categories identified do not appear to be mutually exclusive with significant overlap between the second and third. Perhaps a table would be helpful here.

Authors' response

We have now included a table (table 1) describing the clinical features of the three categories of circulatory impairment.

Summary of findings tables (GRADE)

Please note that these should be updated. The definitions as they pertain to quality of evidence are no longer valid. Evidence is conceptualized as confidence in estimates of effect and not in terms of the likelihood of future research making a difference.

Authors' response

The definitions of GRADE quality of evidence have now been revised to reflect confidence in estimates of effect (Supplementary tables 2 and 4):

High quality: We are very confident that the true effect lies close to that of the estimate of the effect.

Moderate quality: We are moderately confident in the effect estimate: The true effect is likely to be close to the estimate of the effect, but there is a possibility that it is substantially different.

Low quality: Our confidence in the effect estimate is limited: The true effect may be substantially different from the estimate of the effect.

Very low quality: We have very little confidence in the effect estimate: The true effect is likely to be substantially different from the estimate of effect.

Table 4

Foot note 3 is problematic. The basis of the indirectness is unclear i.e. why is the African setting problematic? Also serious indirectness usually implies a drop by two levels.

Authors' response

The main reason for downgrading was because of the indirectness of the patient population – that is, children with severe hypotension were excluded (effects of treatments administered were likely to be substantially different in this higher risk group).

We have now omitted the following statements from the footnote:

'The trial was conducted in hospital settings in Kenya, Tanzania and Uganda. Children were aged 2 months to 12 years. Children with severe malnutrition, gastroenteritis, or shock due to trauma surgery or burns were excluded.'

According to GRADE guidelines serious indirectness implies a drop by one level while very serious indirectness, for example due to problems in more than one PICO (Patient, Intervention, Comparison, Outcome) element suggest a need to rate down the quality of evidence by two levels:

Guyatt GH, Oxman AD, Kunz R, Woodcock J, Brozek J, Helfand M, Alonso-Coello P, Falck-Ytter Y, Jaeschke R, Vist G, Akl EA, Post PN, Norris S, Meerpohl J, Shukla VK, Nasser M, Schünemann HJ; GRADE Working Group. GRADE guidelines: 8. Rating the quality of evidence--indirectness. *J Clin Epidemiol.* 2011 Dec;64(12):1303-10.

GRADE evidence profiles would be useful to include as well to facilitate a better understanding of the limitations of the evidence base.

Authors' response

Agree, we have now included GRADE evidence profiles documenting our judgement of the determinants of quality of evidence for each outcome, in addition to the summary of findings for each outcome (Supplementary tables 3 and 5).